# The Influence of Family Socioeconomic Status on Adolescents’ Mental Health in China

**DOI:** 10.3390/ijerph19137824

**Published:** 2022-06-26

**Authors:** Dongliang Yang, Shuxian Hu, Mingna Li

**Affiliations:** 1Northeast Asian Research Center, Jilin University, Changchun 130012, China; yangdl@jlu.edu.cn; 2School of Northeast Asian, Jilin University, Changchun 130012, China; shuxian_hu@163.com; 3School of Literature, Changchun University, Changchun 130012, China

**Keywords:** family socioeconomic status, education, mental health, Adolescents, heterogeneity

## Abstract

Adolescent mental health is an important public health issue that cannot be ignored, and mental health issues are a major cause of physical illness in adolescents and children worldwide. In order to supplement relevant research and provide insightful policy suggestions to government, schools and families, this study investigated the nexus between family socioeconomic status and mental health for adolescents in China. Based on 7234 observations from the China Education Panel Survey, the paper estimated the effects of family socioeconomic status on adolescents’ mental health using ordinary least squares. The effects of family income, parental education level, and parental occupation on adolescent mental health were estimated separately by the same method. Our findings suggest that family socioeconomic status has a significant positive effect on mental health of adolescents in China. This effect is heterogeneous depending on individuals’ registered residence types and their participation in health courses. Finally, frequency of parent-child interaction and classmate interaction are shown to be potential mechanisms for the effect of family socioeconomic status on the mental health of Chinese adolescents.

## 1. Introduction

With the development of society, the mental health of adolescents is becoming more and more important. Research has shown that 10–20% of children and adolescents suffer from mental health problems, which are a major cause of physical health problems in the group, around the world [1]. However, in low and middle income countries, the mental health of children and adolescents is often overlooked. In recent years, the proportion of adolescents suffering from mental health problems and depression has been increasing. According to the “Report on the Development of Chinese National Mental Health (2019–2020)” released by the Institute of Psychology of the Chinese Academy of Sciences in March 2021, the rates of depression in primary school, junior high school, and high school are 10%, 30%, and 40%, respectively. Among them, the rates of major depression were 1.9–3.3%, 7.6–8.6%, and 10.9–12.6%, respectively. That is, depression levels increased with grades [2]. Therefore, the mental health of adolescents is so important that we cannot ignore it. In order to protect the mental health of adolescents, it is necessary to pay attention to the important role of the family.

Family socioeconomic status which reflects an individual’s family background and social capital, mainly including family income, parents’ education level, etc., is crucial to adolescents’ mental health [3]. To explore the influence of family socioeconomic status on adolescents’ mental health, this paper established an ordinary least squares regression model based on the CEPS from 2014 to 2015. The effects of family income, parental education level, and parental occupation on adolescent mental health were estimated separately by the same method.

Different from the existing research, the marginal contribution of this paper lies in the following three aspects. First, this paper investigated the effect of family socioeconomic status on mental health in a developing country: China. The study used nationally representative data in China to analyze how family socioeconomic status affects adolescents’ mental health. It analyzes the three indicators used to denote family socioeconomic status: family income, parents’ education level, parents’ occupation, and the relationship between these indicators and mental health. This can enrich the literature in the field. Second, in order to explore the influence of family socioeconomic status on adolescents’ mental health more clearly, this study analyzed the heterogeneity of this influence across health course participation status and different registered residence type. Third, we studied two mechanisms by which family socioeconomic status affects mental health: the frequency of parent-child interaction and the frequency of classmate interaction; that is, family socioeconomic status affects adolescents mental health by affecting the frequency of interaction with parents and classmates.

## 2. Literature Review

As the spiritual pillar of physical health, mental health which plays a key role in health is affected by a variety of factors.. The World Health Organization (WHO) defined health in its broader sense in 1946 as “a state of complete physical, mental, and social well-being and not merely the absence of disease or infirmity”. [4] Chen et al. discussed the nature of mental health from the perspective of social adaptation function [5], pointing out that the essence of mental health is the interdependence or balance between the individual in terms of the external environment and the inner heart. Many scholars have studied the influencing factors of mental health from different perspectives. Huang et al. [6] studied the relationship between national income and mental health, and pointed out that national income has heterogeneous effects on different types of mental health. Company-Córdoba et al. [7] found that the mental health of adolescents is related to socioeconomic factors (parents’ education level and family income); that is, children with high income parents and high parental education level have better adaptability. Gunasiri et al. [8] believe that the mental health of young people is related to climate, in that young people feel the negative effects of climate change, manifested as anxiety, ecological anxiety, despair, and stress. In addition, studies have shown that physical exercise also has significant impact on mental health of adolescents. Physical exercise promotes the mental health of adolescents (Bonhauser et al.; Li et al.; Granero-Jiménez et al.) [9,10,11]; that is, physical exercise can alleviate the psychological pressure of teenagers, help them to prevent negative emotions and improve self-esteem.

Although there is a wide literature on the impact on mental health from different perspectives such as national income, socioeconomic factors, climate change, and physical exercise, there are relatively few studies on mental health from the perspective of family socioeconomic status, especially for Chinese teenagers. Moreover, there is no consistent conclusion on the impact mechanism by which family socioeconomic status affects mental health. Link et al. proposed that family socioeconomic status and social support, having a great impact on the health level of individuals in society, are the “basic factors” affecting health [12]. Wang found that there is a phenomenon of health inequality among the Chinese people, which is manifested in that the higher the socioeconomic status, the higher the health level [13]. Xu et al. held that improving family socioeconomic status and parental income is conducive to raising children to be healthy adults [14], because high family socioeconomic status can not only provide adequate nutrition for children’s physical development, but also promote children’s academic and cognitive development [15]. Liu et al. believe that it is the wealth of resources that households with high family socioeconomic status have that increases the well-being of family members [16]. Matthews et al. pointed out that one way to link socioeconomic status and mental health is that individuals with low socioeconomic status lack positive interpersonal relationships and personal internal resources [3], that is, low socioeconomic status lower levels of mental health due to interpersonal and personal resources. In addition, subjective social status and family function mediate the relationship between family socioeconomic status and mental health [17,18]. Jiang et al. (2018) held that parent-child relationships, peer relationships and teacher-student relationships play a complete mediating role in the relationship between socioeconomic status and children’s mental health [19].

The above literature explores the impact of family socioeconomic status on health from different perspectives, and puts forward many constructive insights, providing theoretical material for this article. However, there are inconsistent conclusions about the mechanism of action of family socioeconomic status on mental health. Therefore, on the basis of studying the influence of family socioeconomic status on adolescents’ mental health, this paper explored two mechanisms, namely the frequency of parent-child interaction and the frequency of classmate interaction on the relationship between family socioeconomic status and adolescents’ mental health. Moreover, this paper studied the heterogeneity of the influence of family socioeconomic status on mental health among groups with different health course participation and different registered residence types. Based on the above analysis, this paper proposes the following five hypotheses.

**Hypothesis 1.** 
*Family socioeconomic status has a significant positive effect on adolescent mental health.*


Family socioeconomic status predicts adolescent hope significantly and positively [18]. Families with high family socioeconomic status can provide more social and economic resources to promote the development of adolescents [20], so adolescents with high family socioeconomic status have a clearer orientation for future development and thus have a higher level of hope. Hope, as one of the positive personality traits and personality strengths recognized in the field of positive psychology, is of great significance to the mental health of adolescents [21]. Based on the above analysis, Hypothesis 1 is put forward.

**Hypothesis 2.** 
*Adolescents who have not taken health courses are more likely to be affected by the socioeconomic status of their families.*


Mental health courses can effectively promote individual mental health. Rigabert et al. [22] show that psychological education interventions can effectively reduce the depressive symptoms of patients. Mental health courses based on psychological quality education can not only solve students’ mental health problems, but also train students to become more positive, optimistic, happy and enterprising individuals [23]. Benoit et al. [24] pointed out that positive psychology interventions can improve adolescent life satisfaction, relieve anxiety and depression in this population, and promote a positive attitude. Individuals with positive and optimistic attitudes are less affected by the socioeconomic status of their families. Based on the above analysis, Hypothesis 2 is put forward.

**Hypothesis 3.** 
*There is heterogeneity in the relationship between different registered residence types on family socioeconomic status and mental health of adolescents.*


There are significant differences in mental health status between urban and rural students. Generally speaking, in China, cities are economically developed areas and rural areas are underdeveloped areas. Li et al. [25] pointed out that children in developed areas have higher family socioeconomic status, and local economic development can promote the healthy growth of children. However, children and students in underdeveloped rural areas have lower family socioeconomic status. Children from families with lower socioeconomic status tend to live in poverty, which has a negative impact on children’s mental health [26]. Therefore, family socioeconomic status has different effects on the mental health of adolescents from different urban and rural areas. Based on the above analysis, Hypothesis 3 is put forward.

**Hypothesis 4.** 
*Family socioeconomic status affects the mental health of adolescents by affecting the frequency of parent-child interaction.*


Families with high socioeconomic status pay more attention to comprehensive interaction with their children. A study by Hou et al. found that parents with high socioeconomic status invest more time and energy in their children’s learning and cultural entertainment, which not only improves their children’s academic performance, but also promotes their children’s mental health [27]. Therefore, families with high socioeconomic status can promote the mental health of teenagers by improving parent-child interaction. Based on the above analysis, Hypothesis 4 is put forward.

**Hypothesis 5.** 
*Family socioeconomic status affects mental health by affecting the frequency of interaction between teenagers and classmates.*


Teenagers with high family socioeconomic status can establish a good peer relationship, and a good peer relationship can promote their mental health. Zhang et al. found that family income, an important indicator of family socioeconomic status, had a significant positive impact on social ability in childhood [28]. The higher the family income, the stronger the social adaptability; it is then easier to be accepted by peers and establish a good relationship with them. Moreover, good peer relationships help to reduce the risk of depression and promote the mental health of adolescents [29]. The frequency of interaction with classmates is an important index to measure peer relationship. Based on the above analysis, Hypothesis 5 is put forward.

## 3. Materials and Methods

### 3.1. Study Design

The data used in this study come from the China Education Panel Survey (CEPS) conducted by Renmin University of China [30]. CEPS is a representative large-scale tracking survey project in China which started in 2013.This data takes junior high school students as the starting point of the survey, and plans to conduct follow-up surveys in the 1st, 3rd, 4th, 7th, 8th, 17th, and 27th years after students graduate from junior high school. It conducts a detailed and comprehensive survey of junior high school students, their families, and schools, which makes the data suitable for the research and analysis presented here. In addition, 112 schools and 438 classes were randomly selected from 28 county-level units randomly selected across the country for investigation. All students in the selected class are sampled. The baseline survey surveyed approximately 20,000 students. It has a large sample size, wide coverage, a reasonable questionnaire design, and scientific survey methods, effectively reflecting the development of our society. However, due to the influence of some force majeure factors, the data is only updated to the 2014–2015 school year. Therefore, this paper only used the CEPS from 2014 to 2015.

In this study, we focused on the relationship of family socioeconomic status and the mental health of adolescents. We used data from the 2014–2015 school year. In the follow-up survey in the 2014–2015 school year, the first grade group of junior high school with a sample number of 10,279 in the baseline survey was the main follow-up object. The number of successful students was 9449, that is, the follow-up rate was 91.9%. In addition, there are 471 new students and 830 lost students. Cleaning the sample data and eliminating the rejected answers, outliers and missing data in the sample, we finally obtained 7234 observation results. Using the econometric software Stata version 16 for statistical analysis, we report the mean, standard deviation, minimum and maximum of variables. The ordinary least square method (OLS) was used to investigate the relationship between family socioeconomic status and mental health of adolescents. To test the mechanism, a mediating effect test was used. (We used the SPSS Process plug-in (selection model 4) provided by Hayes [31]). All reported *p*-values were two-tailed.

### 3.2. Variable Descriptions

#### 3.2.1. Explained Variable: Mental Health

The mental health index adopts the mental health scale of the student questionnaire of CEPS, which has been widely used [32,33]. Question C25 in the questionnaire asked, “in the past seven days, do you have the following feelings: ” depressed “,” too depressed to concentrate on work “, ” unhappy “, ” life is meaningless “, ” unable to do things “, ” sad “, ” nervous “, ” premonition that bad things will happen “, ” too much energy, not paying attention in class “, ” worrying too much “, using positive assignment: Assign the option “always”, “often”, “sometimes” rarely” and “never” to 1, 2, 3, 4, 5 at a time. Each question is scored from 0 to 5, and we weighted the scores by entropy weight method to measure mental health. Thus, the variable of mental health ranged from 1 to 5, the higher the value, the better the mental health. Table 1 reports the results of entropy weight assignment.

#### 3.2.2. Explanatory Variable: Family Socioeconomic Status

According to Buchmann [34], the family socioeconomic status that affects the mental health of teenagers is mainly composed of their family income, parents’ education level and parents’ occupation. Following Ren’s research [35], we measured the family socioeconomic status of adolescents. With regard to the measurement of household income, in previous studies, the respondents were often asked to fill in the accurate income directly, but income is private; the interviewees often refused to answer or answered falsely, resulting in a decline in the recovery rate and accuracy of the questionnaire. Moreover, the subjective family income level of teenagers has a greater effect on their mental health than the objective family income. Therefore, this study selects “what do you think of your family’s current financial level” from the students’ questionnaire to measure family income, and sets the option to “very difficult”, “more difficult”, “medium”, “relatively rich” and “very rich”. The option is scored from 1 to 5, the higher the score, the higher the level of family income.

The educational level of parents can be obtained by asking parents about their academic qualifications. Parents’ academic qualifications are divided into primary school and below, junior high school/technical secondary school/technical school, vocational high school/ordinary high school, college, undergraduate and above. The option is scored from 1 to 5, the higher the score, the higher the parents’ education.

With regard to parental occupation, refer to the “Occupational Classification Code of China (2015 version)” proposed by the National Bureau of Statistics in 2015 [36]. In this paper, occupations are divided into the following five categories: unemployed/laid-off, ordinary workers/farmers/herdsmen/fishermen/primary workers, staff/business, service personnel/skilled workers/individual industrial and commercial households, doctors/lawyers/primary and secondary school teachers/accountants/nurses/software programmers and other technical staff, government leaders/cadres/institutions/companies (enterprises) leaders/cadres/scientists/engineers/university teachers and other professional and technical personnel. The option is scored from 1 to 5.

The socioeconomic status of the family is composed of three indicators: family income, parents’ education level and parents’ occupation, in which the education level and occupation of the parents are measured by the highest party of both parents. The present study used the entropy weight method to synthesize the family income, parental education level, and parents’ occupation into a family socioeconomic status index which is a continuous variable of 1–5. The higher the index score, the higher the family socioeconomic status. Among them, the weights of family income, education level and parents’ occupation are 0.222, 0.421 and 0.357 according to standard deviation method.

#### 3.2.3. Control Variables

The main activities of teenagers are distributed in two places: school and family. Generally, the influencing factors of teenagers’ mental health mainly come from three aspects: school, family and individual characteristics. The main control variables in this paper are registered residence type, confidence, unhealthy diet, parent-child relationship, parent-child communication, parental strictness, parents’ active contact with teachers, school’s atmosphere, health course, positive effect of good friends, parent-child interaction frequency, classmates’ interaction frequency. The meaning and descriptive statistics of the variables used are shown in Table 2.

### 3.3. Empirical Methodologies

In order to accurately estimate the effect of family socioeconomic status on the mental health of adolescents, the ordinary least square method was used to construct the model. The model for building the model is as follows.
(1)$${\mathrm{mental}}_{i}=\alpha_{0}+\beta_{0}\mathrm{fss}+{\gamma \mathrm{control}}_{i}+\varepsilon_{i}$$
(2)$${\mathrm{mental}}_{i}=\alpha_{0}+\beta_{0}\mathrm{income}+{\gamma \mathrm{control}}_{i}+\varepsilon_{i}$$
(3)$${\mathrm{mental}}_{i}=\alpha_{0}+\beta_{0}\mathrm{education}+{\gamma \mathrm{control}}_{i}+\varepsilon_{i}$$
(4)$${\mathrm{mental}}_{i}=\alpha_{0}+\beta_{0}\mathrm{occupation}+{\gamma \mathrm{control}}_{i}+\varepsilon_{i}$$

Models (1)–(4) estimates the effects of family socioeconomic status, family income, parents’ education level and parents’ work on the mental health of adolescents. Among them, ${\mathrm{mental}}_{i}$ is the explained variable, which represents the mental health of the individual. Fss, income, education and occupation are the explanatory variables, indicating the family socioeconomic status, family income, parents’ education level and parents’ occupation respectively. ${\mathrm{control}}_{i}$ is the control variable, which controls other factors affecting the mental health of adolescents. $\varepsilon_{i}$ is a random error term.

### 3.4. Empirical Results

There was no problem of collinearity in the models (see Appendix A for the results of the collinearity test).

Table 3 shows the OLS results for the effects of family socioeconomic status, family income, parents’ education level and parents’ occupation on adolescents’ mental health. Specifically, column 2 reports the regression results of family socioeconomic status to mental health, column 3 reports the regression results of family income to mental health, column 4 reports the regression results of parents’ education level to adolescents’ mental health, and column 5 reports the regression results of parents’ occupation to adolescents’ mental health. As indicated in column 2, family socioeconomic status (0.043) has a statistically significant effect on the mental health of adolescents at 5% level, and the higher the family socioeconomic status, the better is the mental health status of adolescents. This conclusion validates Hypothesis 1. The results in columns 3 and 4 show that family income (0.099) has significant positive effects on adolescents’ mental health at 1% level, and parents’ education level (0.026) have significant positive effects on adolescents’ mental health at 5% level. The results in column 5 show that parental occupation (0.004) has no significant effect on the mental health of adolescents. By comparing the data of the students’ questionnaire and the parents’ questionnaire provided by CEPS, we find that 6122 of the 10,750 students do not know their parents’ occupation. That is to say, nearly 60% of adolescents cannot clearly state their parents’ occupations, which means that parents’ occupations have no direct effect on adolescents’ mental health.

Our model shows that when controlling a group of variables, family socioeconomic status, family income and parents’ education level are important factors affecting adolescents’ mental health. There is a significant positive correlation between them and the mental health of adolescents. From the point of view of the coefficient, the effect of family income (0.099) on the mental health of teenagers is greater than that of parents’ education level (0.026). Moreover, the coefficients of unhealthy diet (−0.216), parent-child relationship (0.076), parents’ active contact with teachers (−0.017), and school atmosphere (0.198) show a statistically significant effect on adolescents’ mental health at 1% level and the positive effects of health courses (0.061); good friends (0.054) have a statistically significant effect on adolescents’ mental health at 5% level.

### 3.5. Robustness Check

#### 3.5.1. Refactoring Explained Variables

Considering the complexity of individual mental health, this part tests the robustness of the model by reconstructing the explained variables. Specifically, the data in the psychological questionnaire is standardized, each question is constructed as data with a mean of 0 and a standard deviation of 1, and finally the data is summed and averaged to be a new explained variable. 

As shown in Column 2, 3 and 4 of Table 4, family socioeconomic status (0.036) and parents’ education level (0.020) still have a statistically significant effect on adolescent mental health at the 5% level, and family income (0.091) still has a statistically significant effect on mental health. As indicated in Column 5, parental occupation (0.002) has no significant effect on adolescent mental health, which is consistent with our previous findings.

By reconstructing the explained variables in this model, we showed that the effects of family socioeconomic status, family income, and parental education level on adolescent mental health are robust. Table 4 reports the robustness test results of the reconstructed explanatory variables.

#### 3.5.2. Winsorization

Large samples may have observations that affect coefficient estimates and inferred extreme values. Two widely used approaches to address influential observations are winsorization and truncation [37]. Here, we use winsorization to test the robustness of the model. In specific, after winsorizing explained variables at 1% level, this variable was added to the previous OLS model.

Table 5 shows the robustness test results for winsorization. As indicated in Column 2, family socioeconomic status (0.045) still has a statistically significant effect on adolescent mental health at the 5% level. As shown in Column 3 and 4, family income (0.099) and parental education level (0.026) still have statistically significant effects on adolescents’ mental health. As indicated in Column 5, parental occupation (0.005) has no significant effect on adolescent mental health, which is consistent with our previous findings.

According to the results of the robustness test of winsorization, the results of the positive effect of family socioeconomic status on adolescents’ mental health are robust.

### 3.6. Heterogeneity

According to whether the individual has studied health courses, we divided the samples into “yes” and “no”, and the sample sizes were 4676 and 2558, respectively. As indicated in Column 2 and 3 of Table 6, the mental health status of adolescents who have not participated in health courses (0.055) is more likely to be affected by family socioeconomic status than individuals who have participated in health courses (0.038). This result confirms Hypothesis 2. Moreover, as shown in Column 4, 5, 6 and 7, the mental health status of adolescents who have not participated in health courses is more affected by family income (0.138) and parents’ education level (0.034). The results in the column 8 and 9 show that parental occupation has no significant effect on the mental health of adolescents, which is consistent with our previous findings.

To sum up, whether or not to take health courses has a heterogeneous effect on the mental health of adolescents. Specifically, compared with the individuals who did not take part in a health course, the mental health status of the individuals who participated in the health course was less affected by the family economic status, family income and parents’ education level.

In China, rural areas are significantly different from urban areas in terms of economy, culture, etc. Although the previous analysis did not find that urban location has a significant effect on individual mental health, there may be heterogeneity for the effect of family socioeconomic status on mental health. According to the individual’s registered residence type, we divided the sample into a rural group and an urban group, with sample sizes of 3857 and 3377, respectively.

As indicated in columns 2 and 3 of Table 7, family socioeconomic status significantly affects mental health of both rural (0.056) and urban (0.036) individuals, the effect on rural individuals is greater than on urban individuals. This result confirms Hypothesis 3. In addition, as shown in columns 4, 5, 6, and 7 of Table 7, compared with urban adolescents, rural adolescents’ mental health is more affected by family income (0.105) and parents’ education level (0.040). The results in columns 8 and 9 show that parental occupation has no significant effect on the mental health of adolescents, which is consistent with our previous findings.

### 3.7. Mechanisms

In order to explore the mechanism behind the effect of family socioeconomic status on adolescent mental health, this section studies two paths: parent-child interaction frequency and classmate interaction frequency.

First of all, we discuss whether there is a mediating effect of parent-child interaction frequency on the effect of family socioeconomic status on adolescents’ mental health. Moreover, whether there is a mediating effect of parent-child interaction frequency on family income and of parents’ education level on teenagers’ mental health is also discussed here. The Bootstrap method of deviation correction is used to test the mediating effect, and the sample size is 5000. 

Table 8 reports the mediating effect test of parent-child interaction frequency. Model (1) in Table 8 shows the mediating effect of parent-child interaction frequency on the effect of family socioeconomic status on adolescent mental health, while model (2) and model (3) show the mediating effect of parent-child interaction frequency on family income and of parents’ education level on adolescent mental health, respectively. As shown in model (1) of Table 8, the 95% confidence interval of the estimated effect for the parent-child interaction frequency is [LLCI = 0.038, ULCI = 0.056], and it does not contain 0, indicating the existence of a mediating effect in the effect of family socioeconomic status on adolescent mental health. What is more, the value of the mediating effect is 0.047, and the proportion of mediating effect in the total effect is 58.02%. This result verifies Hypothesis 4. Similarly, as shown in Table 8, model (2) and Table 8, model (3), the 95% confidence intervals of the estimated mediation effects are, respectively, [0.044,0.067] and [0.036,0.060]; they do not contain 0, indicating that the frequency of parent-child interaction has a mediating effect on the relationship between family income, parents’ education level and adolescents’ mental health.

Then, on the basis of controlling variables, we use the same method as above to explore whether there is a mediating effect of classmate interaction frequency in the effect of family socioeconomic status on adolescents’ mental health.

The test results are shown in Table 9. Model (1) reports the mediating effect of social frequency on the relationship between family socioeconomic status and mental health of adolescents. Under 95% confidence interval of deviation correction, interval [LLCI = 0.012, ULCI = 0.029] does not contain 0, indicating that social frequency has a significant mediating role in the effect of family socioeconomic status on adolescent mental health. The mediating effect is 0.016, accounting for 19.75% of the total effect. This result confirms Hypothesis 5. In addition, model (2) and model (3) in Table 9 report the mediating effect of social frequency on the effect of family income and parents’ education level on the mental health of adolescents, respectively.

## 4. Discussion

Based on the CEPS data from 2014 to 2015, this study uses the entropy weight method to synthesize the family socioeconomic status from three indicators: family income, parents’ education level and parents’ occupation, and uses the psychological scale in the students’ questionnaire as an index to measure the mental health of adolescents. First of all, this paper establishes a multiple linear regression model to analyze the effect of family socioeconomic status on adolescents’ mental health, and analyzes the effects of family income, parents’ education level and parents’ occupation on teenagers’ mental health, respectively. We found that family socioeconomic status, family income and parents’ education level have statistically significant effects on adolescent mental health. These results are robust. Then, we study the heterogeneity of the effect on mental health, and analyze the effect of family socioeconomic status on the mental health of adolescents in the case of different exposure to health courses and different types of household registration. Finally, the frequency of parent-child interaction and the frequency of interaction with classmates are used as intermediary variables to analyze the mechanism of the effect of family socioeconomic status on the mental health of adolescents.

Family socioeconomic status has a significant positive effect on mental health, which is consistent with previous conclusions [17]. Family socioeconomic status is a comprehensive multi-dimensional concept, which is mainly measured by family income, parents’ education level, parents’ occupation and other factors [34]. The higher the socioeconomic status of the family, the more social resources the family members have access to, and the higher the family income, education level and occupation. Higher family income can ensure children’s nutritional intake and promote their growth into healthy adults. Parents with high education and professional levels pay more attention to their children’s mental health and pay more attention to parenting styles and high-quality parent-child interaction. Therefore, higher family income, parents’ education level and parents’ occupation have a positive effect on teenagers’ mental health, and the improvement of family socioeconomic status can promote teenagers’ mental health.

Family socioeconomic status has different effects on the mental health of groups with different health course exposures and registered residence types. Compared with individuals who have not taken health courses, the mental health status of individuals who have taken health courses is less affected by family socioeconomic status, which is consistent with the conclusion that mental health courses have a significant positive effect on mental health [22,23,24]. Health courses can improve students’ health awareness and psychological quality, and promote the establishment of a healthy life attitude, positive psychological state and correct outlook on life. According to the economic differences between urban and rural areas in China, the family socioeconomic status of urban residents is relatively high, while that of rural residents is relatively low. Young people living in cities can enjoy various forms of entertainment, rich educational resources and other resources that are lacking in rural areas. Due to the huge differences in living conditions and social resources, there are also differences in the mental health status of urban and rural adolescents [25,26]. When the socioeconomic status of the family changes, the mental health of rural teenagers will be more affected.

Cheng et al. [38] proposed that psychological quality is an effective link between family socioeconomic status and mental health of middle school students. However, this paper puts forward the intermediary role of parent-child interaction frequency and classmate interaction frequency between family socioeconomic status and adolescent mental health. For individuals with higher family socioeconomic status, on the one hand, the higher their parents’ awareness of mental health, they more they pay attention to high-quality and multiple parent-child companionship to promote their children’s mental health growth, On the other hand, when parents can use more money and social resources to train and educate their children, this improves their children’s social adaptability, and children with strong social adaptation can establish a good relationship with their classmates (good peer relationships show up as a high frequency of classmate interaction) [28,29], thus promoting mental health. Therefore, the frequency of parent-child interaction and classmate interaction play an intermediary role in the relationship between family socioeconomic status and adolescent mental health.

However, this paper is limited in some facets. First, we only estimate the short-term effect of family socioeconomic status on adolescent mental health. Regretfully, due to data constraints, we fail to take into account the dynamic effect of family socioeconomic status on the mental health of adolescents. Due to the influence of some force majeure factors, the data is only updated to the 2014–2015 school year. We are unable to analyze current adolescent mental health problems, even if mental health problems among youths have dramatically increased during the COVID 19 pandemic. Second, this study only uses the indicators of the psychological scale in the CEPS students’ questionnaire to construct mental health indicators. If more indicators can be obtained, the measurement of mental health will be more accurate. Third, this paper only focuses on the frequency of parent-child interaction and classmate interaction, ignoring other possible influencing mechanisms. To this end, we will refine the research in both data and theoretical frameworks to more carefully assess the effect of family socioeconomic status on adolescent mental health and explore new mechanisms.

## 5. Conclusions

Based on the CEPS data from 2014 to 2015, this paper studies the effect of family socioeconomic status on the mental health of adolescents by using the ordinary least square method, and makes a robustness test and heterogeneity analysis. We found that family socioeconomic status, family income and parents’ education level have significant positive effects on teenagers’ mental health, and the heterogeneity of these effects depends on whether individuals have studied health courses and on individual types of household registration. Specifically, the mental health of individuals who have not studied health courses and who have agricultural registered residence are more affected by the socioeconomic status of the family. In addition, we point out two mechanisms of the effect of family socioeconomic status on teenagers’ mental health: the frequency of parent-child interaction and the frequency of classmate interaction. On the one hand, improving the socioeconomic status of the family can improve the frequency of parent-child interaction and then improve the mental health level of adolescents; on the other hand, the socioeconomic status of the family can affect the frequency of interaction between individuals and classmates, affect peer relationships and then affect the mental health of adolescents.

Based on the above research, this paper puts forward the following three suggestions. First, the government should focus on improving the family socioeconomic status of residents. The government can improve the socioeconomic status of residents’ families by increasing their disposable income, improving residents’ awareness of lifelong learning, and increasing employment opportunities, thereby promoting the mental health of young people. Second, schools should strengthen health education for young people. All primary and secondary schools should be fully aware of the importance and necessity of health curriculum construction, and strengthen health education for teenagers by offering health courses and holding health lectures, so as to help teenagers establish correct health awareness and improve their psychological quality. Third, families should improve the frequency and quality of parent-child interaction. Parents should balance the relationship between family and work, raise their awareness of parent-child interaction, spend more quality time with their children, and increase the frequency of parent-child interaction so as to improve the health level of teenagers.

## Figures and Tables

**Table 1 ijerph-19-07824-t001:** Entropy weight result.

In the Past Seven Days, Have you Had the Following Feelings	Index Variability	Index Conflict	Information Content	Weight
depression	1.056	4.062	4.289	0.09
too depressed to concentrate on work	1.094	3.976	4.349	0.091
unhappy	1.074	3.803	4.085	0.086
life is meaningless	1.102	4.087	4.505	0.095
unable to do things	1.084	3.985	4.318	0.091
sad	1.059	3.823	4.049	0.085
nervous	1.067	4.698	5.015	0.105
premonition that bad things will happen	1.183	4.644	5.496	0.116
too much energy, not paying attention in class	1.102	6.033	6.649	0.14
worrying too much	1.099	4.373	4.804	0.101

Note: Based on CEPS from 2014 to 2015.

**Table 2 ijerph-19-07824-t002:** Variables and descriptive Statistics.

Variable	Description	Mean	SD	Min	Max
Mental health	Adolescents’ mental health	3.830	0.799	1	5
Fss	Family socioeconomic status of adolescents, which is composed of three aspects: family income, parents’ education level and parents’ occupation.	2.695	0.766	1	5
Income	Family income, which is from the question “what do you think of your family’s current financial level?”	2.957	0.599	1	5
Education	Parents’ education level (data from parents’ questionnaire)	2.718	1.152	1	5
Occupation	Parents’ occupation (data come from the parents’ questionnaire)	2.506	0.977	1	5
Type	Registered residence type	0.467	0.499	0	1
Confidence	What do you think of your appearance?	3.081	0.664	1	5
Diet	Unhealthy diet	2.779	0.728	1	5
Relationship	Parent-child relationship	2.616	0.452	1	3
Communication	Parent-child communication	2.230	0.536	1	3
Strictness	Parental strictness	2.315	0.394	1	3
Contact	Parents’ active contact with teachers	2.345	1.012	1	4
Atmosphere	School atmosphere: “most of the students in my class are very friendly to me.” my class has a good atmosphere.” “ I often take part in the work organized by the school or the class.”	3.063	0.678	1	4
Course	Health course: Did you take a health education class when you were in primary school?	0.646	0.478	0	1
Effect	Positive effect of good friends	2.456	0.521	1	3
Classmate	Classmate interaction frequency	2.288	1.237	1	6
Parent-child	Parent-child interaction frequency	2.296	1.141	1	6

Note: Based on CEPS from 2014 to 2015.

**Table 3 ijerph-19-07824-t003:** OLS regression results of mental health of adolescents.

Variables	Model (1)	Model (2)	Model (3)	Model (4)
Fss	0.043 ** (0.013)			
Income		0.099 *** (0.015)		
Education			0.026 ** (0.009)	
Occupation				0.004 (0.009)
Type	−0.005 (0.020)	0.003 (0.018)	−0.004 (0.020)	0.021 (0.018)
Confidence	0.133 *** (0.013)	0.124 *** (0.013)	0.135 *** (0.013)	0.137 *** (0.013)
Diet	−0.216 *** (0.012)	−0.222 *** (0.012)	−0.215 *** (0.012)	−0.214 *** (0.012)
Relationship	0.276 *** (0.021)	0.272 *** (0.021)	0.277 *** (0.021)	0.276 *** (0.021)
Communication	0.015 (0.019)	0.0153 (0.019)	0.160 (0.019)	0.025 (0.019)
Strictness	−0.046 (0.025)	−0.045 (0.025)	−0.046 (0.025)	−0.050 (0.025)
Contact	−0.017 *** (0.009)	−0.015 (0.009)	−0.016 (0.009)	−0.015 (0.009)
Atmosphere	0.198 *** (0.014)	0.193 *** (0.014)	0.199 *** (0.014)	0.199 *** (0.014)
Course	0.061 ** (0.019)	0.056 ** (0.019)	0.061 ** (0.017)	0.063 ** (0.019)
Effect	0.054 ** (0.018)	0.054 ** (0.018)	0.056 ** (0.018)	0.061 ** (0.018)
_cons	2.523 *** (0.091)	2.399 *** (0.093)	2.547 *** (0.091)	2.534 *** (0.091)
$\mathrm{Adj}\;R^{2}$	0.150	0.153	0.149	0.148

Note: *** *p* < 0.01, ** *p* < 0.05.

**Table 4 ijerph-19-07824-t004:** Robustness test table for reconstructed explanatory variables.

Variables	Model (1)	Model (2)	Model (3)	Model (4)
Fss	0.036 ** (0.012)			
Income		0.091 *** (0.014)		
Education			0.020 ** (0.008)	
Occupation				0.002 (0.009)
Control	YES	YES	YES	YES
_cons	−1.197 *** (0.085)	−1.313 *** (0.087)	−1.178 *** (0.085)	−1.187 *** (0.085)
N	7234	7234	7234	7234
$\mathrm{Adj}\;R^{2}$	0.147	0.151	0.147	0.146

Note: *** *p* < 0.01, ** *p* < 0.05.

**Table 5 ijerph-19-07824-t005:** Robustness test table for winsorization.

Variables	Model (1)	Model (2)	Model (3)	Model (4)
Fss	0.045 ** (0.013)			
Income		0.099 *** (0.015)		
Education			0.026 *** (0.009)	
Occupation				0.005 (0.009)
Control	YES	YES	YES	YES
_cons	2.530 *** (0.090)	2.406 *** (0.093)	2.555 *** (0.090)	2.541 *** (0.090)
N	7234	7234	7234	7234
$\mathrm{Adj}\;R^{2}$	0.150	0.154	0.150	0.149

NOTE: *** *p* < 0.01, ** *p* < 0.05.

**Table 6 ijerph-19-07824-t006:** The heterogeneous effect of the Health Course.

Variable	Model (1)		Model (2)		Model (3)		Model (4)	
	YES	NO	YES	NO	YES	NO	YES	NO
Fss	0.038 **	0.055 **						
	(0.016)	(0.024)						
Income			0.073 *** (0.019)	0.138 *** (0.026)				
Education					0.022 ** (0.011)	0.034 ** (0.016)		
Occupation							0.009 (0.011)	−0.008 (0.017)
Control	YES	YES	YES	YES	YES	YES	YES	YES
_cons	2.600 ***	2.488 ***	2.514 **	2.289 ***	2.618 ***	2.526 ***	2.607 ***	2.518 ***
	(0.118)	(0.150)	(0.120)	(0.155)	(0.118)	(0.150)	(0.118)	(0.150)
N	4676	2558	4676	2558	4676	2558	4676	2558
$\mathrm{Adj}\;R^{2}$	0.154	0.121	0.156	0.129	0.154	0.121	0.153	0.119

Note: *** *p* < 0.01, ** *p* < 0.05.

**Table 7 ijerph-19-07824-t007:** Heterogeneity of registered residence type.

Variable	Model (1)		Model (2)		Model (3)		Model (4)	
	Rural	Urban	Rural	Urban	Rural	Urban	Rural	Urban
Fss	0.056 **	0.036 **						
	(0.024)	(0.017)						
Income			0.105 *** (0.020)	0.083 ** (0.024)				
Education					0.040 ** (0.016)	0.019 (0.011)		
Occupation							−0.023 (0.016)	0.132 (0.012)
Control	YES	YES	YES	YES	YES	YES	YES	YES
_cons	2.552 ***	2.457 ***	2.461 ***	2.347 ***	2.576 ***	2.479 ***	2.664 ***	2.477 ***
	(0.123)	(0.141)	(0.122)	(0.146)	(0.120)	(0.140)	(0.120)	(0.141)
N	3857	3377	3857	3377	3857	3377	3857	3377
$\mathrm{Adj}\;R^{2}$	0.138	0.162	0.143	0.164	0.138	0.162	0.137	0.161

Note: *** *p* < 0.01, ** *p* < 0.05.

**Table 8 ijerph-19-07824-t008:** Mediating effect Test Table of parent-child interaction frequency.

	**Model (1) Explanatory Variable: Family Socioeconomic Status**				
	**Effect Value**	**SE**	**LLCI**	**ULCI**	**Ratio of Effect**
Mediating effect	0.047	0.005	0.038	0.056	58.02%
Direct effect	0.034	0.011	0.014	0.055	41.98%
Total effect	0.081	0.010	0.062	0.100	
	**Model (2) Explanatory Variable: Family Income**				
	**Effect Value**	**SE**	**LLCI**	**ULCI**	**Ratio of Effect**
Mediating effect	0.055	0.006	0.044	0.067	33.95%
Direct effect	0.107	0.017	0.074	0.139	66.05%
Total effect	0.162	0.016	0.131	0.192	
	**Model (3) Explanatory Variable: Parents’ Education Level**				
	**Effect Value**	**SE**	**LLCI**	**ULCI**	**Ratio of Effect**
Mediating effect	0.035	0.003	0.036	0.060	61.40%
Direct effect	0.022	0.009	0.005	0.039	38.60%
Total effect	0.057	0.008	0.041	0.073	

**Table 9 ijerph-19-07824-t009:** Mediating Effect Test Table of Social Frequency.

	**Model (1) Explanatory Variable: Family Socioeconomic Status**				
	**Effect Value**	**SE**	**LLCI**	**ULCI**	**Ratio of Effect**
Mediating effect	0.016	0.004	0.012	0.029	19.75%
Direct effect	0.065	0.010	0.045	0.085	80.25%
Total effect	0.081	0.010	0.062	0.100	
	**Model (2) Explanatory Variable: Family Income**				
	**Effect Value**	**SE**	**LLCI**	**ULCI**	**Ratio of Effect**
Mediating effect	0.021	0.005	0.015	0.038	12.96%
Direct effect	0.141	0.016	0.109	0.172	87.04%
Total effect	0.162	0.016	0.131	0.192	
	**Model (3) Explanatory Variable: Parents’ Education Level**				
	**Effect Value**	**SE**	**LLCI**	**ULCI**	**Ratio of Effect**
Mediating effect	0.013	0.003	0.008	0.018	22.81%
Direct effect	0.044	0.008	0.027	0.060	77.19%
Total effect	0.057	0.008	0.041	0.073	

## Data Availability

The CEPS data can be accessed through its official website, http://ceps.ruc.edu.cn/ (accessed on 10 May 2022).

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
