# Peer review of "The Influence of Family Socioeconomic Status on Adolescents’ Mental Health in China"

_ijerph, 2022, doi:10.3390/ijerph19137824_

Round 1

Reviewer 1 Report

The study:

The mental health of adolescents, as the authors state, is indeed a most relevant public health concern. Despite being an ancient concern, it has never cessed to grow. Most of the times, research in psychology tends to approach adolescents mental health from an individual perspective. This study, however, falls into another category as it approaches the subject from an ecological point of view, namely, addressing the impact of the family's socioeconomic status on adolescents' mental health. Such an approach allows to tackle issues of social inequality and to frame adolescents' mental health in a public perspective: not as a purely "psychoilogical" issue, individually understood, but as a broader sociopolitical issue that must be addressed in terms of public health, taking into account social, political, and economic factors. In doing so, the present study helps to address adolescents' mental health issues from a holistic and ecological perspective, pointing to socioeconomic, cultural and political factors that necessarily have to be taken into consideration. Yet, as the authors point out, it is also extremely important to capture the way(s) socioeconomic status (SES) impacts mental health. This article has the merit to address two possible relevant mechanisms by which SES may impact adolescents' mental health: frequency of parent-child inter-action and the frequency of classmate interaction. Also the role of hope, positivity and optimism learned in health classes is explored.

Method:

In terms of method, it is relevant to underline the sample's N (N=7234) from 112 schools and 438 classes, which adds to the robustness of the statistical analyses. Furthermore, it is part of an ambitious study that foresees 7 waves of data collection, allowing for a projected extensive longitudinal design. The statistical tests used are appropriate for testing the research hypotheses, including the mediating effect of parent-child interaction frequency and classmate interaction frequency on the influence of family socioeconomic status (calculated from 3 indicators: family income, parents’ education level and parents’ occupation) on adolescent mental health.

Discussion and conclusion:

The direct and indirect effects of the family SES on the adolescents’ mental health are appropriately explored. Family resources associated with SES (financial, educational, cultural capital) are proven to be a crucial determinant of adolescents’ mental health. The study has, thus, the merit of pointing to the fact that psychological outcomes (in this case, mental health) are a product of ecological conditions and resources (in the present case, the family’s SES). The study’s results adds to other studies that stress the relevance of material, social and cultural conditions to the overall well-being of individuals, pointing to the fact that mental health is not solely an individual concern but a political issue that asks for a comprehensive socioeconomic approach.

Author Response

Thank you for your valuable comment. Adolescent mental health is an important public health issue. In order to supplement relevant research and provide insightful policy suggestions to government, schools and families, we investigated the nexus between family socioeconomic status and mental health for adolescents in China. Our findings suggest that family socioeconomic status has a significant positive effect on mental health of adolescents in China. Frequency of parent-child interaction and classmate interaction are shown to be the potential mechanism through which the effect of family socioeconomic status on mental health of Chinese adolescents. As you said, it is true that mental health is not solely an individual concern but a political issue. Thanks again.

Reviewer 2 Report

It is a well-conducted study, of current relevance that lays the foundation for future studies on the socio-emotional and psychosocial functioning of adolescents.

The only technical observation is for the consideration of the authors. Although a level of statistical significance can indeed be established at p < 0.1, it is unusual. The vast majority of the significant results presented in each table have the usual statistical significance (p < .01 - .05). I suggest reconsidering such meanings so as not to discern from the rest of the literature.

Author Response

Thank you for your valuable comment. Following your suggestion, we deleted *p<0.1 in Table3,4,5,6,7. Only few parameters in table 3 are significant at p<0.1, so we corrected them too. The revisions were marked in the text.

Reviewer 3 Report

The paper entitled “The Influence of Family Socioeconomic Status on Adolescents' Mental Health in China” was focused on an important issue, as the mental health problems have dramatically increased during pandemic COVID 19 among youths. Thus the discussed subject is both important and current. 

The advantages of the manuscript are the large study sample, the justification for the study hypothesis, and the adequate statistical analysis.

The main controversy is the data from 2014-15, which is quite old and is not relevant to the current population, especially after COVID 19 experience. Thus I suggest to add this information to study limitation.

There are also unclear descriptions in the methodological section that must be better described e.g. The way of describing parents' educational level or parental occupation, after the list of academic qualifications or occupations the authors stated that participants rated it from 1 to 5 – I actually do not understand what for and why, as these are categorical options and either you have, for example, higher or elementary education, so what was rated?

Next the “a multiple linear regression model” does not allow to formulate causal conclusions, and the words “impact”, and “influence” indicate such a relationship – must be corrected

Additional small editorial errors should be corrected:

-the subsection about the statistical analysis and statistical program is recommended to be added

-The minor editorial errors throughout the whole paper - citations inconsistent with the journal's guidelines (e.g. line 86), lack of spaces (e.g. line 126-7, etc.) or doubled spaces (e.g. line419); the title of table 7 at the end of the page (line 379

Author Response

Thank you for the opportunity to revise our manuscript entitled “The Influence of Family Socioeconomic Status on Adolescents' Mental Health in China” (ID: ijerph- 1751972). We have studied the comments carefully and try our best to make corresponding corrections. The point-by-point responses to the reviewer’s comments are enclosed. To show the revisions more clearly, we have marked all corresponding revisions in blue.
